

# Lithospheric and sub-lithospheric deformation under the Borborema Province of NE Brazil from receiver function harmonic stripping

Gaelle Lamarque[1,2] and Jordi Julià[1,3]

[1]Programa do Pós-Graduação em Geodinâmica e Geofísica, Universidade Federal do Rio Grande do Norte, Natal, RN CEP 59078-090, Brazil
[2]Ifremer, Geosciences Marines, Centre de Brest, 29280 Plouzané, France
[3]Departamento de Geofísica, Universidade Federal do Rio Grande do Norte, Natal, RN CEP 59078-970, Brazil

**Correspondence:** Gaelle Lamarque (gaelle.lamarque@ifremer.fr)

**Abstract.** The depth-dependent anisotropic structure of the lithosphere under the Borborema Province of northeast Brazil has been investigated through harmonic stripping of receiver functions developed at 39 stations in the region. This method retrieves the first (k=1) and second (k=2) degree harmonics of a receiver function dataset, which characterize seismic anisotropy beneath

a seismic station. Anisotropic fabrics are in turn directly related to the deformation of the lithosphere from past and current tectonic processes. Our results reveal the presence of anisotropy within the crust and the lithospheric mantle throughout the entire Province, with the exception of a few stations in the continental interior that lack evidence for any anisotropic signatures. Most stations in the continental interior report consistent anisotropic orientations in the crust and lithospheric mantle, suggesting a dominant NE-SW pervasive deformation along lithospheric-scale shear zones developed during the Brasiliano-Pan African

orogeny. The lack of anisotropy at a few stations along a NE-SW trend in the center on the Province is harder to explain, but might be related to heating of the lithosphere by an asthenospheric channel. Finally, several stations along the Atlantic coast reveal depth-dependent anisotropic orientations roughly (sub)perpendicular to the margin. These results suggest a more recent overprint, probably related to the presence of frozen anisotropy in the lithosphere due to stretching and rifting during the opening of the South Atlantic.

*Copyright statement.*

## 1  Introduction

Understanding intraplate deformation and its relationship to deep geodynamic processes such as sublithospheric flow is critical for improving our understanding of the evolution of continents. The Borborema Province of NE Brazil, for instance, has witnessed several cycles of deformation, as well as recurrent episodes of intraplate volcanism and uplift, during its geological

history. Brasiliano-Pan African deformation is well represented through the network of shear zones that pervade the Borborema



Province (Vauchez et al., 1995; Neves et al., 2000). These shear zones separate several tectonic terrains of Paleoproterozoic and Archean age that amalgamated and/or were reworked during the orogeny (Jardim de Sá et al., 1992; Cordani et al., 2003). Major Neoproterozoic shear zones thus constitute inherited structures that could have influenced the geometry of subsequent tectonic processes, such as the opening of the South Atlantic Ocean (Tommasi and Vauchez, 2001; Kirkpatrick et al., 2013).

Also, current topography of the Borborema Plateau and the Sertaneja Depression may have resulted from a combination of on-going deep processes, such as edge-driven convection in the asthenospheric mantle and/or stretching and thinning of the lithosphere during Mesozoic times (de Oliveira and Medeiros, 2012; Almeida et al., 2015).

Recent seismological studies from receiver functions (Almeida et al., 2015; Pinheiro and Julia, 2014; Luz et al., 2015a, b), ambient noise (Dias et al., 2014) or P-wave tomography (Simões Neto et al., 2019), and SKS splitting (Bastow et al., 2015)

have greatly contributed to further our understanding of the relationships between inherited Precambrian structures, Mesozoic extensional processes, and episodes of post-breakup volcanism and uplift. However, several tectonic and geodynamic questions remain unanswered. In particular, the presence of the Meso-Cenozoic Macau-Queimadas volcanism (MQA, figure 1) - which does not present a clear age progression - remains unclear. Moreover, SKS-waves showed little to no evidence of splitting in the continental interior (Bastow et al., 2015), which is difficult to comprehend given the complex tectonic and deformational

history of the Province.

Here, we determine depth-dependent anisotropy in the Borborema lithosphere (crust and mantle) from harmonic analysis of receiver functions. Our results confirm that SKS splitting at coastal stations is dominated by fossil anisotropic fabrics in the lithospheric mantle, likely originating from Mezosoic extension. In the continental interior, receiver function stripping reveal fast-axis orientations consistent with major regional shear zones, suggesting their continuation at depth into the lithospheric

mantle. Our results also confirm the absence of lithospheric (fossil) anisotropy under stations that did not record SKS-splitting. These stations are aligned along a NE-SW trend located west and north of the Borborema Plateau, a high-standing topographic feature that rises  ∼1000 m above mean sea level. We argue that the absence of anisotropy in the lithosphere is related to sub-lihospheric heating of the overlying lithosphere by a shallow asthenospheric channel under the Province.

## 2   Geological setting

The Borborema Province formed during the Neoproterozoic Braziliano/Pan-African orogeny (600-580 Ma), as a result of the collision between the São Luiz-West Africa craton to the north and the São Francisco-Congo craton to the South (Jardim de Sá et al., 1992; Cordani et al., 2003). It thus represents the west central portion of a larger Neoproterozoic belt that resulted from the assembly of the Gondwana supercontinent.

The basement of the Borborema Province comprises mostly gneisses and migmatitic rocks of Paleoproterozoic age, and

small Archean nuclei, overlain by Neoproterozoic metasediments formed during the Brasiliano orogeny (Neves, 2003). This basement is affected by an extensive network of Neoproterozoic shear zones oriented EW and NE-SW (Figure 1). These shear zones are major structures several hundreds of kilometers long and tens of kilometers wide (Vauchez and da Silva, 1992) that can be traced into the African continent in paleogeographic reconstructions (Arthaud et al., 2008). The Borborema shear





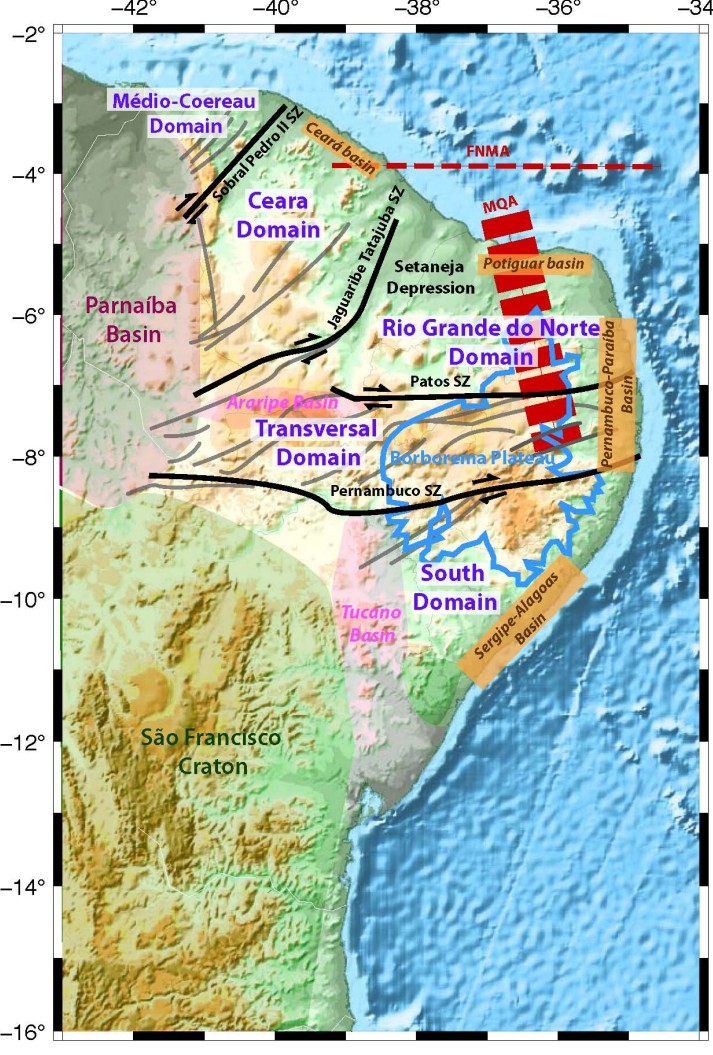

**Figure 1.** Topographic map of northeast Brazil with main geological features of the Borborema Province superimposed. Black and grey lines correspond to major shear zones (SZ) and red dashed lines to the volcanic alignments of Fernando de Noronha-Mecejana (FNMA) and Macau-Qeimada (MQA). The Borborema Plateau boundaries are indicated in blue.

zones were activated in high temperature and high-to-low pressure conditions and are associated with a strong production of magmas from both crustal and mantle sources (Vauchez et al., 1995). The shear zone network can be split into two domains: a western domain of NE-striking skike-slip faults, and an eastern domain of more sinuous, discontinuous E-W-striking shear zones (Vauchez et al., 1995). These two domains could be related to two discrete collisional events with the Parnaíba block to the west and the São Francisco craton to the south, respectively, which forced NE extrusion of the Province at the end of the Neoproterozoic (Araújo et al., 2014).




The geodynamic evolution of this basement and the significance of these shear zones is still debated, and two main models have been traditionally proposed. On one hand, the accretionary model proposes that the Borborema Province comprises of several Paleoproterozoic small continental fragments that aggregated along the shear zones, which then constitute lithospheric-scale suture zones separating independent tectonic blocks (Cordani et al., 2003; Van Schmus et al., 2011; Araújo et al., 2014).

The number of independent terrains is unclear, but there is a general consensus in arranging them into five major Precambrian domains: (i) Médio-Coereau, in the northwestern most tip of the Province; (ii) Ceará, between the Sobral-Pedro II and Jaguaribe-Tatajuba shear zones; (iii) Rio Grande do Norte, immediately east of the Ceará domain; (iv) Transversal or Central, between the Patos and Pernambuco lineaments; and (v) Southern, immediately north of the São Francisco craton (Figure 1). On the other hand, it has been suggested that the Borborema Province was a single unit since 2.0 Ga and that the shear zones

recorded intracontinental supracrustal deformation during the Brasiliano orogeny (Tommasi et al., 1995; Vauchez et al., 1995; Neves, 2003). In this later model, micro-plate amalgamation largely predates the Brasiliano orogeny, which would have only partially reworked preexisting plate structures in the Borborema Province.

During the Mezosoic, opening of the south Atlantic separated the Borborema Province from its African conjugate (de Matos, 1992). Continental rifting resulted in significant crustal thinning in the region (Santos et al., 2014; Lima Neto et al., 2013;

Luz et al., 2015b), forming both marginal (e.g. Ceará, Potiguar, Pernambuco-Paraíba, Sergipe-Alagoas) and intra-continental (e.g. Araripe, Tucano) sedimentary basins (Figure 1). The post Gondwana breakup evolution of the Borborema Province is characterized by recurrent magmatism (Knesel et al., 2011) and postulated episodes of uplift in the Borborema Plateau (Morais Neto et al., 2009; de Oliveira and Medeiros, 2012) and the Araripe basin (Assine, 2007; Marques et al., 2014). Intraplate volcanism is characterized as small-volume, long-lived and mainly alkalic in nature (Knesel et al., 2011). It is

arranged along two main linear alignments of Mesozoic-Cenozoic volcanic rocks (Figure 1): the Macau-Queimadas Alignment (MQA), mostly on-shore and approximately trending in the north-south direction; and the Fernando de Noronha-Mecejana Alignment (FNMA), mostly off-shore and trending in the east-west direction (Mizusaki et al., 2002; Knesel et al., 2011, and references therein). The MQA displays K/Ar ages ranging from 80 to 30 Ma (Mizusaki et al., 2002) and 40Ar/39Ar ranging from 93 to 7 Ma (Knesel et al., 2011) without a clear age progression, whereas the FNMA displays progressive K/Ar and

40Ar/39Ar ages from the Fernando de Noronha archipelago (22 to 2 Ma) to the west (Knesel et al., 2011, and references therein), to the Mecejana volcanism (34 to 26 Ma) to the east (Mizusaki et al., 2002).

Cenozoic uplift was inferred from both relative dating of elevated sediments of the Serra dos Martins formation in the northern Borborema Province and absolute dating from apatite fission-track analysis of granitic-gneissic samples (Morais Neto et al., 2009; de Oliveira and Medeiros, 2012). Although some sort of tectonic uplift and/or inversion of the Araripe basin

seems to be widely accepted (Marques et al., 2014; Peulvast and Bétard, 2015; Garcia et al., 2019), uplift in the Borborema Plateau is more debated. On one hand, de Oliveira and Medeiros (2012) argue that this tectonic event is linked to Cenozoic mafic underplating and isostatic uplift due to a small-scale convection cell at the edge of the continent, which might have also been responsible for the surface volcanism (Knesel et al., 2011). The hypothesis of a thin layer of mafic underplate seems to be consistent with recent receiver functions observations south of the Patos Lineament (Luz et al., 2015b, a). However,

Luz et al. (2015b) debated the time of emplacement of such a mafic cumulates. These authors proposed that this mafic layer



would be part of the original Proterozoic crust, and that the southern Borborema Plateau should be regarded as a high-standing, rheologically strong block surrounded by stretched and delaminated crust. The stretching model seems to have been confirmed by a rheological contrast along the Patos lineament postulated from seismic P-wave tomography (Simões Neto et al., 2019). The tomographic study also identifies an asthenospheric low-velocity channel trending NE-SW under the center of the Province,

which is interpreted as resulting from lateral flow from a distant mantle plume. Such asthenospheric flow might represent the source of Meso-Cenozoic intraplate volcanism in NE Brazil.

## 3   Data and methodology

### 3.1   Seismic data

Seismic data for this study were obtained at 75 seismic stations in northeast Brazil. These stations belong to a variety of

seismic networks, both permanent and temporary. The Rede Sismográfica do Nordeste (RSISNE) consists of 19 broadband stations equipped with RefTek 151-120 sensors feeding RT-130 digitizers (24-bit) sampling at 100 Hz, with an inter-station spacing of about 250 km and a network aperture of ~800 km. The RSISNE network has been in operation since 2011 and was initially funded by the national oil company Petrobras. The Instituto Nacional de Ciência e Tecnologia em Estudos Tectônicos (INCT-ET) network, consists of 7 broadband stations and 22 short-period stations. The broadband stations were arranged along

an approximately 1000 km-long line with interstation spacing of about 100 km. They were equipped with STS-2.5 Streckheisen sensors and Q330 data loggers (24-bit) sampling at 100 Hz. The 22 short-period stations were equipped with L4A-3D Sercel sensors (2 Hz cut-off frequency) and 24-bit RT-130 digitizers sampling at 100 Hz. They were in operations between 2011 and 2012 and recorded continuously between 6 months and 1.5 years. The INCT-ET network was funded by the Conselho Nacional de Desenvolvimento Científico e Tecnológico (CNPq). Up to 6 broadband stations operated during 2007–2009 under

the Institutos do Milênio project. These stations were equipped with KS2000 Geotech sensors and Geotech digitizers sampling continuously at 100 Hz. They recorded continuously for periods ranging from 6 months to 2 years, with two of them still in operation. This network was also funded by CNPq. Station RCBR belongs to the Global Seismographic Network (GSN). This station has been recording since 03/1999 with a CMG-3T Guralp sensor, which was replaced in 07/2004 by a STS-2 Streckeisen sensor, always feeding a Q330 data logger and sampling continuously at 40 Hz. Seven broadband stations belonging to the

broader Brazilian Lithosphere Seismic Project (BLSP) (Assumpção et al., 2004) operated for 1.5 to 3.0 years in NE Brazil. They were equipped with either CMG-3T Guralp or STS-2 Streckeisen sensors and 24-bit RT-130 digitizers sampling at 100 Hz. Finally, 11 broadband stations deployed under the BOrborema Deep Electromagnetic and Seismic (BODES) experiment were installed along an approximately NS line crossing the Araripe basin. They were equipped with RefTek 151-120 sensors and RT-130 digitizers. The stations were in operation between 2015 and 2017 and recorded continuously for ~2 years at 100

Hz. Further detail on the 75 seismic stations is given in the Table S1 and their geographical location displayed in Figure 2.

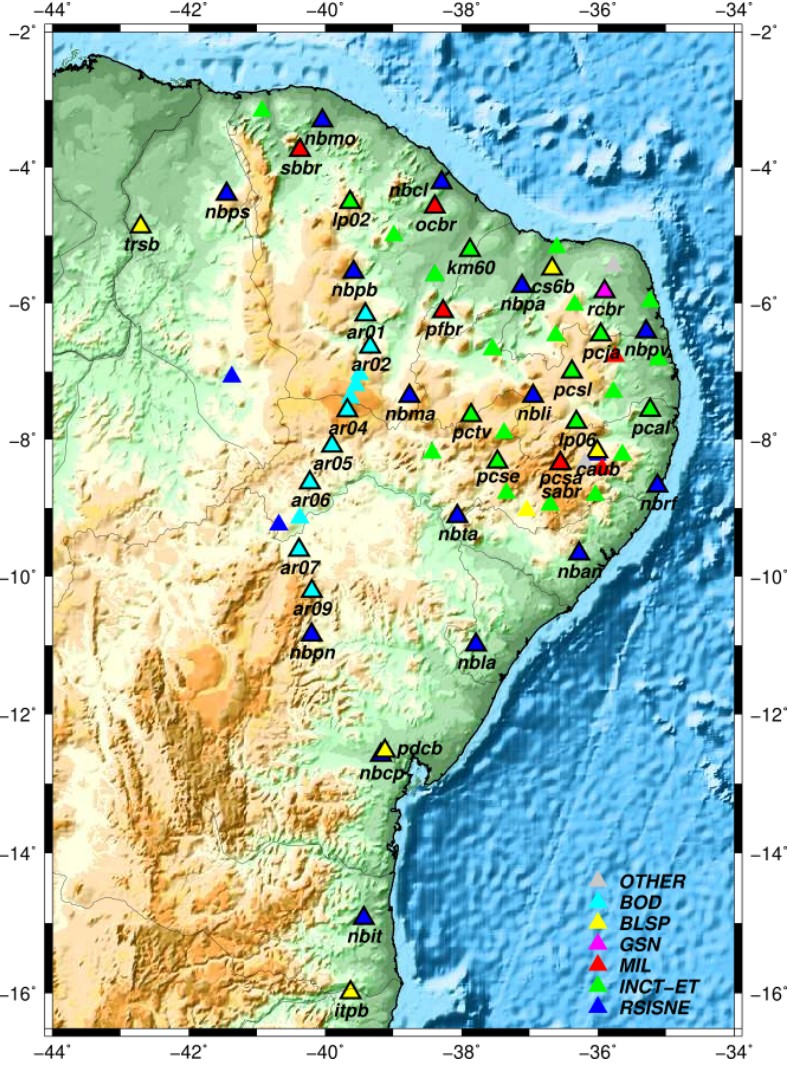

**Figure 2.** Topographic map of northeast Brazil with locations of broadband and short-period stations considered in this study. Stations were color-coded by network: stations from the network RSISNE are represented in dark blue, INCT-ET in green, Milenio in red, GSN in pink, BLSP in yellow, Bodes in light blue and others networks in grey (see legend). Only the selected stations have been named and are represented with black contour.

## 3.2 Receiver function processing and migration

Receiver functions were computed for the 75 stations making the combined network for northeast Brazil. Most of the receiver function estimates were developed by Luz et al. (2015a) in order to investigate lateral variations in crustal thickness and bulk Vp/Vs ratio across the Borborema Province. This dataset was later utilized by Almeida et al. (2015) to study the crustal architecture of the region from receiver function Common Conversion Point (CCP) stacks. We developed 1400 new receiver



functions estimates at 11 temporary stations from the BODES network, following the same procedure as Luz et al. (2015a). The receiver function approach aims at retrieving P-to-S converted phases within the coda of teleseismic P waves that result from the interaction of the teleseismic P-wavefront with crustal and upper mantle discontinuities under the recording station (Langston, 1979, 1977) in order to produce estimates of the depth of the discontinuities. The converted phases are detected

by deconvolving the radial and tangential components of the teleseismic waveforms by the corresponding vertical component (Ammon, 1991). This operation removes the effects of the source time function, near source propagation and instrumental response from the seismograms, leaving the signature of propagation local to the receiver.

The main processing steps involved in the development of the receiver function estimates are summarized below, and further details about computational and quality control procedures can be found in Luz et al. (2015a). First, we selected seismic sources

with magnitude greater than 5.0 mb and occurring at epicentral distances between 30° and 90° from the selected stations (see Figure 3). The corresponding waveforms were then windowed 10 s before and 110 s after the P-wave arrival time, demeaned, detrended, tapered with a 5% cosine taper, and high-pass filtered above 0.05 Hz to remove low-frequency noise. All waveforms were re-sampled to 20 Hz, after low-pass filtering below 8 Hz to avoid aliasing. Before deconvolution, the waveforms were additionally low-pass filtered below 1.25 Hz with an acausal Gaussian filter (Gaussian width 2.5). The deconvolution procedure

of the vertical component from the radial and transverse components was implemented through the iterative, time-domain procedure of Ligorria and Ammon (1999), with 500 iterations. The deconvolved time series were again filtered with the same Gaussian filter of width 2.5. Percent recoveries of the observed radial component under 85% were automatically rejected and the remaining receiver functions were visually inspected for each station to identify and remove outliers.

Prior to implementing the anisotropy analysis, each radial and tangential receiver function was migrated to depth after P to

S ray-tracing through the global velocity model ak135-f (Kennett et al., 1995; Montagner and Kennett, 1996). The purpose of the migration is to correct the phase move-out introduced by varying incidence angles among the incoming teleseismic P-wavefronts, effectively equalizing the receiver function waveforms in the depth domain (Dueker and Sheehan, 1997). Next, the migrated radial and transverse receiver functions for each station were grouped by back-azimuth in 36 non-overlapping, 10° wide bins, and averaged within each bin. A given station was then selected if it presented ~~at least~~ two averaged receiver

functions (one radial and one tangential) in at least 9 bins. This selection criterion ensured a sampling of at least 90° in back-azimuth, either continuously or discontinuously, around the station. A total of 39 stations were thus selected for anisotropy analysis. An example of stacked and migrated receiver functions is displayed in Figure 4.

## 3.3  Estimating depth-dependent anisotropy within the lithosphere

In order to map deformation within the lithosphere, we estimate seismic anisotropy from the harmonic decomposition of

receiver functions. The harmonic stripping method is described in Shiomi and Park (2008); Bianchi et al. (2010); Audet (2015). The method assumes that, at every depth, an ensemble of receiver functions can be expressed as a linear combination of $\cos(k\phi)$ and $\sin(k\phi)$ terms, where $k$ is the harmonic degree or order, and $\phi$ is the back-azimuth. Shiomi and Park (2008) show that, for anisotropic media, radial and tangential receiver functions display a $\pi/2k$ shift for both $k = 1$ and $k = 2$ harmonic degrees; the tangential receiver functions can thus be added to the radial component after applying a phase shift of $+\pi/2k$





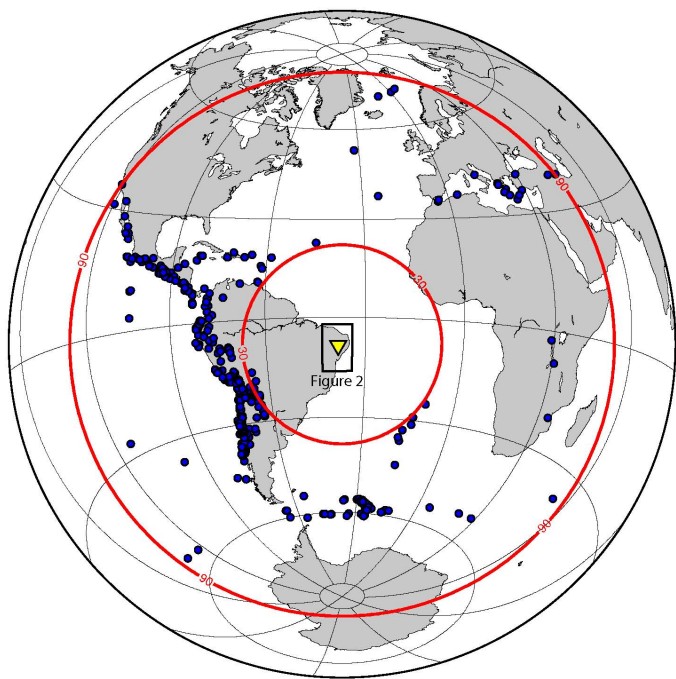

**Figure 3.** Location of earthquakes (blue circles) used for receiver function analyses, occurring at epicentral distances between 30°and 90°(red lines) and with magnitude $Mb \geq 5.0$.. The yellow triangle corresponds to the seismic network presented in Figure 2.

and naturally improve the azimuthal coverage around the station. After the harmonic decomposition is performed, up to 5 coefficient functions, corresponding to the first three harmonics, are obtained ($k = 0, 1, 2$). The first harmonic ($k = 0$) represents the isotropic variations from flat interfaces in an equivalent isotropic medium; for this harmonic, the signal is only presents in the radial component. If anisotropic structures are present at depth, the second and third harmonics ($k = 1$ and $k = 2$) contain

5    energy with periodicity of $2\pi/k$. For $k = 1$, a two-lobed periodicity of $2\pi$ is either related to the presence of a dipping interface or to an anisotropic layer with a plunging symmetry axis (Maupin and Park, 2007). Two coefficient functions express the projection of this harmonic along the N-S and E-W directions, which correspond to the coefficients multiplying the $\cos(\phi)$ and $\sin(\phi)$ terms, respectively. For $k = 2$, a four-lobed periodicity of $\pi$ is related to the presence of an anisotropic layer with a horizontal symmetry axis (Maupin and Park, 2007). As for the second degree harmonic, two coefficient functions express

10   the projection of this harmonic degree along the N-S and 45°N directions, corresponding to the coefficients multiplying the $\cos(2\phi)$ and $\sin(2\phi)$ terms, respectively. The harmonic decomposition can be expressed in matrix form (eq. 1) and solved for the 5 coefficients for the 3 harmonic degrees ($k = 0, 1, 2$) through a singular value decomposition. These harmonics are calculated for every depth within a selected depth-window.





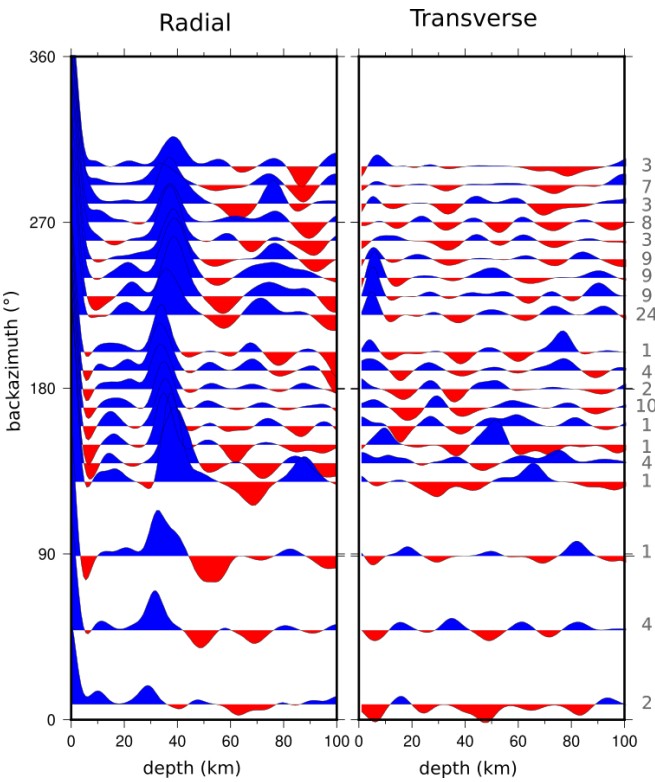

**Figure 4.** Example of stacked receiver functions represented by backazimuth bins of 10°at station PFBR. Grey numbers correspond to the number of stacked receiver functions.)

The matrix equation that implements the harmonic decomposition is given by

$$
\begin{bmatrix}
R_1(z) \\
\vdots \\
R_n(z) \\
T_1(z) \\
\vdots \\
T_n(z)
\end{bmatrix}
=
\begin{bmatrix}
1 & \cos(\phi_1) & \sin(\phi_1) & \cos(2\phi_1) & \sin(2\phi_1) \\
\vdots & \vdots & \vdots & \vdots & \vdots \\
1 & \cos(\phi_n) & \sin(\phi_n) & \cos(2\phi_n) & \sin(2\phi_n) \\
0 & \cos(\phi_1 + \pi/2) & \sin(\phi_1 + \pi/2) & \cos(2(\phi_1 + \pi/4)) & \sin(2(\phi_1 + \pi/4)) \\
\vdots & \vdots & \vdots & \vdots & \vdots \\
0 & \cos(\phi_n + \pi/2) & \sin(\phi_n + \pi/2) & \cos(2(\phi_n + \pi/4)) & \sin(2(\phi_n + \pi/4))
\end{bmatrix}
*
\begin{bmatrix}
A(z) \\
B(z) \\
C(z) \\
D(z) \\
E(z)
\end{bmatrix}
\tag{1}
$$

where $\phi_i$ is the back-azimuth of the i-th R (radial) and T (tangential) receiver function doublet, $A(z)$ represents the first harmonic coefficient ($k = 0$), $B(z)$ and $C(z)$ are the coefficient functions of the second harmonic ($k = 1$), and $D(z)$ and $E(z)$ are the coefficient functions of the third ($k = 2$) harmonic.

5    After solving the matrix equation (1) within a specific depth-window and calculating the five harmonic coefficients, we search for the presence of anisotropy by inspecting the $B(z), C(z), D(z)$ and/or $E(z)$ terms. If at least one of these component





displays non-zero amplitudes, we calculate the energy of the second ($k = 1$) and the third ($k = 2$) harmonic degrees as proposed by Licciardi and Piana Agostinetti (2016):

$$E_{k=1} = \sum_{z=1}^{n}(B(z)^2 + C(z)^2) \tag{2}$$

and

$$E_{k=2} = \sum_{z=1}^{n}(D(z)^2 + E(z)^2) \tag{3}$$

These energy functions allow to discriminate between dipping interfaces/a plunging axis of symmetry and horizontal anisotropy. If $E_{k=1} > E_{k=2}$, the dominant anisotropy is either a dipping interface or an anisotropic layer with a plunging axis of symmetry. In that case, we rotate $B(z)$ and $C(z)$ in discrete back-azimuth increments $\alpha$ (where $\alpha \in [0,2]$) and search for the value of $\alpha$ that maximizes $B(z)$ (and therefore minimizes $C(z)$). This value of $\alpha$ can be directly interpreted as either the trend of the dip, in the case of dipping interface, or as the trend of the fast axis of symmetry in the case of plunging axis of symmetry. If $E_{k=2} > E_{k=1}$, the dominant anisotropy is an anisotropic layer with a horizontal axis of symmetry. In that case, we rotate $D(z)$ and $E(z)$ for each angle increment $\alpha$ (where $\alpha \in [0,2]$) and search for the value of $\alpha$ that maximizes $D(z)$ (and therefore minimizes $E(z)$). This value of $\alpha$ can be directly interpreted as the trend of either the fast or the slow axis of symmetry.

An example of harmonic decomposition is shown for station PFBR in Figure 5. In order to estimate uncertainties, we applied a bootstrap statistical approach by randomly re-sampling with replacement our receiver functions. We performed such analysis with 200 replications at each of the selected stations. A measurement is considered as not reliable, and then rejected, if the estimated uncertainties are larger than 20°.

## 4  Results

Anisotropy parameters were examined for each station at two depth-window ranges: (1) crust (Figure 6A), which was assumed to be located between 0 and 33 km depth, in agreement with the 32-40 km range estimated by Luz et al. (2015b) under the Borborema Plateau and 30-33 km under the surrounding basins; and (2) lithospheric mantle, which was taken to be between 33 and 100 km depth (Figure 6B). All results are indicated in Table 1.

An inspection of Figure 6A reveals that the crust of northeast Brazil presents seismic anisotropy, both within the interior of the continent and along the coast. A number of stations, however, display uncertainties larger than 20°. The significance of these large uncertainties are discussed in section 5. Unresolved anisotropic directions within the crust are recorded around longitude -40° for stations nbpb, ar02, ar05, ar06, nbpn, at the border of the Borborema Plateau (stations nbta, pctv, nbli, caub), and within the Sergipe-Alagoas and Pernambuco basins (stations nban and pcal). The majority of stations that sample clear anisotropic directions display a NE-SW to E-W trending axis of symmetry, except stations cs6b (trend $\sim$ NNW-SSE), km60 and nbma (trends N-S). We mainly measure anisotropy with $2\pi$-periodicity (k=1) related to a dipping interface or anisotropy





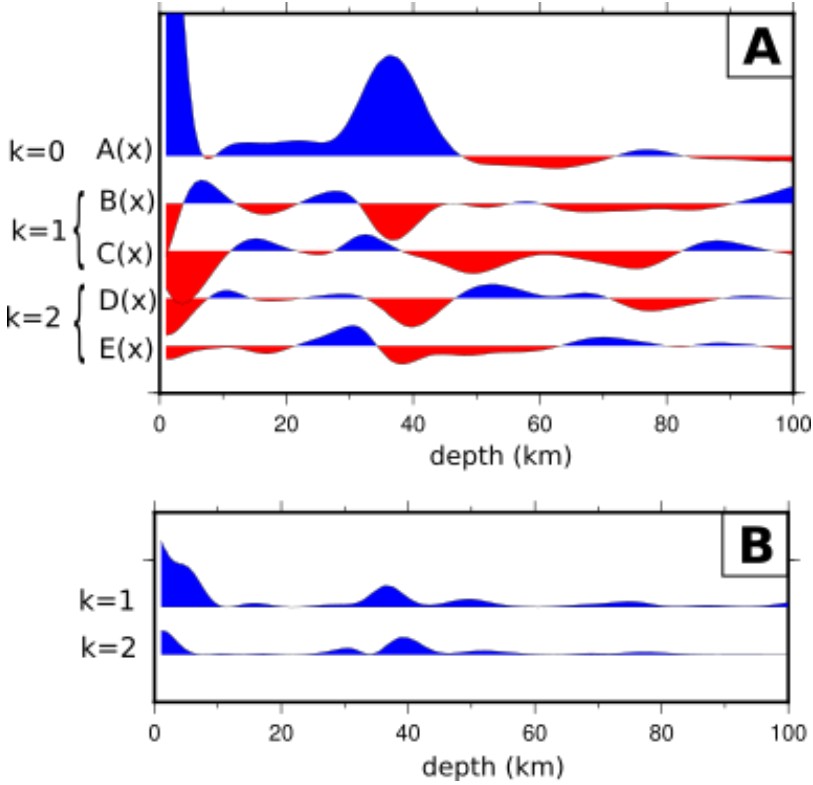

**Figure 5.** Example of results obtained at station PFBR with the harmonic stripping method of Bianchi et al. (2010). A/ From top to bottom are represented the harmonic functions obtained by solving the equation 1: $A(z)$ (first harmonic degree ($k=0$), $B(z)$ ($\cos(\phi)$ term of the second degree harmonic ($k=1$)), $C(z)$ ($\sin(\phi)$ term of the second degree harmonic ($k=1$), $D(z)$ ($\cos(2\phi)$ term of the third ($k=2$) harmonics), and, $E(z)$ ($\sin(2\phi)$ term of the third ($k=2$) harmonics. B/ The energy is represented for harmonic degrees k=1 and k=2

with a plunging axis of symmetry, but some stations display clear $\pi$-periodic horizontal anisotropy (stations ar09, sabr, pcsa, pcsl, pcja, lp06, nbmo). Realize that, in most cases, even though k=1 harmonics display higher energy contents than k=2 harmonics, both pairs of harmonics display energies with comparable strengths (see supplementary materials, Figure S1). For example, station PFBR (Figure 5) shows clear, non-zero energy levels for both k=1 and k=2 harmonics in the crust between 0

5    and 33 km.

Figure 6B shows that the lithospheric mantle is characterized by seismic anisotropy throughout the entire Province, with the exception of a few stations that display large uncertainties (discussed in section 5). Those include stations within the Parnaíba basin (trsb), around longitude -40° (ar01, ao05 and nbpn), along the southern portion of the Borborema Plateau (nbta and pcse), and along a NE-SW axis located northwest of the Borborema Plateau (nbma, pfbr, nbpa, cs6b). Most anisotropic directions trend NE-SW to E-W, with the exception of stations km60 and nbma that show N-S trends. As for the crust, we mainly measure

10   anisotropy with $2\pi$-periodicity (k=1) related to a dipping interface or anisotropy with a plunging axis of symmetry, but some stations display clear $\pi$-periodic horizontal anisotropy (stations ar02, km60, lp06, pcal, nban, nbmo, nbpb, pctv, rcbr). Note





that stations located along the continental margin show anisotropy within the lithospheric mantle with a fast axis of symmetry that is oblique (stations nbmo, nbpv, nbrf, nban, nbit) or perpendicular (stations nbcl, pcal) to the coast.

In the case where stations recorded anisotropic directions at both crustal and mantellic levels, most of them show consistent orientations in the two domains. There are a few instances, nonetheless, in which unaligned orientations for the crust and the
5    lithospheric mantle are observed (ar04, ar09, nbcl, nbit, nbps, nbrf and rcbr).

**Figure 6.** A/ Map of symmetry directions (dark sticks) obtained for the crust (0-32 km). When one stick is plotted at the station, it represents either the trend of the dip in the case of dipping interface or the trend of the fast axis in the case of plunging anisotropy. When two lines are plotted, they refer to the fast axis and to its perpendicular direction in the case of horizontal anisotropy. Light colors represent uncertainties estimated from the bootstrap quantification (200 re-sampling per station). B/ Same for the lithospheric mantle (32-100 km).



**Table 1.** Results of anisotropic symmetry directions for several depths ranges: 0-32 km, 32-100 km and 0-100 km. One direction corresponds either to the trend of the dip in the case of dipping interface or to the trend of the fast axis in the case of plunging anisotropy. When two directions are indicated, they refer to the fast axis and to its perpendicular direction (horizontal anisotropy). Uncertainties were estimated from bootstrap quantification (200 re-sampling at every station).

| Station | 0-32 km | 32-100 km | Station | 0-32 km | 32-100 km |
|---------|---------|-----------|---------|---------|-----------|
| ar01 | $40 \pm 8$ | $44 \pm 20$ | nbpa | $70 \pm 8$ | - |
| ar02 | - | $83.5\text{-}173.5 \pm 7$ | nbpb | - | $84.5\text{-}174.5 \pm 12$ |
| ar04 | - | $47.5\text{-}137.5 \pm 6$ | nbpn | $83 \pm 8$ | $14.5\text{-}104.5 \pm 8$ |
| ar05 | - | $89\text{-}179 \pm 11$ | nbps | $13 \pm 10$ | $60 \pm 20$ |
| ar06 | - | $30 \pm 10$ | nbpv | $40 \pm 5$ | $48 \pm 9$ |
| ar07 | $36 \pm 6$ | $59 \pm 10$ | nbrf | $49 \pm 6$ | $82 \pm 9$ |
| ar09 | $66\text{-}156 \pm 5$ | $33 \pm 14$ | nbta | $0\text{-}90 \pm 16$ | - |
| caub | - | - | ocbr | $74 \pm 19$ | - |
| cs6b | $155 \pm 16$ | $144 \pm 17$ | pcal | $150 \pm 19$ | $84.5\text{-}174.5 \pm 5$ |
| itpb | $19 \pm 5$ | $71 \pm 16$ | pcja | $40 \pm 19$ | $54 \pm 5$ |
| km60 | $10 \pm 16$ | $2.5\text{-}92.5 \pm 5$ | pcsa | $77\text{-}167 \pm 5$ | $74 \pm 10$ |
| lp02 | $109 \pm 20$ | - | pcse | - | $61.5\text{-}151.5 \pm 10$ |
| lp06 | - | $63\text{-}163 \pm 6$ | pcsl | $53\text{-}143 \pm 5$ | $54 \pm 10$ |
| nban | - | $10\text{-}100 \pm 17$ | pctv | - | - |
| nbcl | $84 \pm 18$ | $51 \pm 8$ | pdcb | - | $64 \pm 10$ |
| nbcp | $48 \pm 5$ | $56 \pm 6$ | pfbr | $94 \pm 9$ | - |
| nbit | $105 \pm 10$ | $47 \pm 12$ | rcbr | $60 \pm 15$ | $15\text{-}105 \pm 5$ |
| nbla | $88 \pm 6$ | $16\text{-}106 \pm 16$ | sabr | - | $44.5\text{-}134.5 \pm 9$ |
| nbli | - | $49 \pm 18$ | sbbr | $82 \pm 5$ | $96 \pm 14$ |
| nbma | $6 \pm 15$ | - | trsb | $103 \pm 20$ | $79\text{-}169 \pm 9$ |
| nbmo | $52\text{-}142 \pm 6$ | $56.5\text{-}146.5 \pm 7$ | | | |

## 5 Discussion

### 5.1 Pervasive anisotropy with (sub)horizontal fast axis of symmetry

As described is section 4, we observe in northeast Brazil a dominance of $2\pi$-periodicity (k=1) anisotropy in the lithospheric mantle, which represents either a dipping interface or anisotropy with a plunging axis of symmetry. However, a close inspection of the energy of the k=2 harmonics also suggests an important contribution from anisotropy with a horizontal axis of symmetry. We were able to replicate this pattern with synthetic receiver functions by assuming anisotropy with a slightly (10 to 15°) dipping axis of symmetry (see supplementary materials, Figure S2). Note that, in that case, both k=1 and k=2 harmonics




display consistent orientations. This is, the orientation inferred from the k=1 harmonic degree is always parallel to one of the two orientations inferred from the k=2 harmonic degree.

Moreover, we noticed that the anisotropic fast axes of symmetry throughout the crust are consistent with those throughout the lithosphere for most of the stations, suggesting a prolongation of crustal structures within the lithospheric mantle. Within the

continental interior, the anisotropic orientations are parallel or sub-parallel to the main E-W to NE-SW shear zone directions (stations ar02, nbli or lp06, for example). The consistency of the fast axis direction of lithospheric anisotropy with large structures observed at the surface suggest a continuation of the main shear zones into the lithospheric mantle, as suggested by Vauchez et al. (2012). A few exceptions, nonetheless, are observed for example at stations sabr, sbbr, and nbpb. Such discrepancies in the anisotropy orientations could be related to more local features such as fluid content, presence of cracks or

plutonic bodies along the shear zones, fractures or mineral assemblages (Levin and Park, 1997; Mainprice and Nicolas, 1989).

### 5.2 Anisotropy along the passive margin

Inspection of stations located along the eastern and equatorial margins reveals that anisotropy exhibits - on average - directions either perpendicular or oblique to the coast in the lithospheric mantle. This observation is in agreement with SKS splitting measurements in this area performed by Bastow et al. (2011); Assumpção et al. (2011). These authors concluded that the

anisotropy reported from SKS splitting along the northeastern Brazilian margins must be related to fossil anisotropy inherited from the opening of the South Atlantic ocean. This interpretation is based on the relatively small time delay measured along the coast.

We compare the independent SKS splitting measurements with our results from harmonic stripping of receiver functions in Figure 7. For a better comparison we chose to represent the k=2 harmonics at stations where SKS splitting were measured

because: (i) we expect only horizontal (recorded on the k=2 harmonics) or slightly dipping anisotropy in such geodynamical context; (ii) we observe in our data that k=1 and k=2 harmonics display energy within the same order of magnitude suggesting slightly dipping to horizontal anisotropy beneath most stations; and (iii) SKS waves are mainly sensible to (sub)horizontal anisotropy (Levin et al., 2007). Figure 7 shows a good agreement between anisotropic orientations recorded by receiver functions and SKS-splitting along the eastern and equatorial margins, confirming that the recorded anisotropy beneath coastal

stations is mainly located in the lithospheric mantle. The oblique to parallel orientation of anisotropy along the east and equatorial coasts, respectively, is consistent with the opening trend of the margin (Moulin et al., 2010).

### 5.3 Asthenospheric flow heating the lithosphere

At a number of stations (ar05, nbma, pfbr, nbpa, cs6b), uncertainties for the direction of the fast axis of anisotropy are larger than 20°. We think that anisotropy is just too small to be confidently retrieved, and interpret those stations as sampling an isotropic

lithosphere. Interestingly, those stations seem to form a remarkable line trending NE-SW that approximately coincides with the location of the Cariri-Potiguar trend. Stations nbta and pcse also seem to align along the same direction more to the East.

One explanation for the absence of lithospheric-scale anisotropy could be the destruction of anisotropic fabrics through sub-lithospheric heating of the overlying lithosphere. This hypothesis was proposed for the Cameroon Volcanic Line (CVL) by



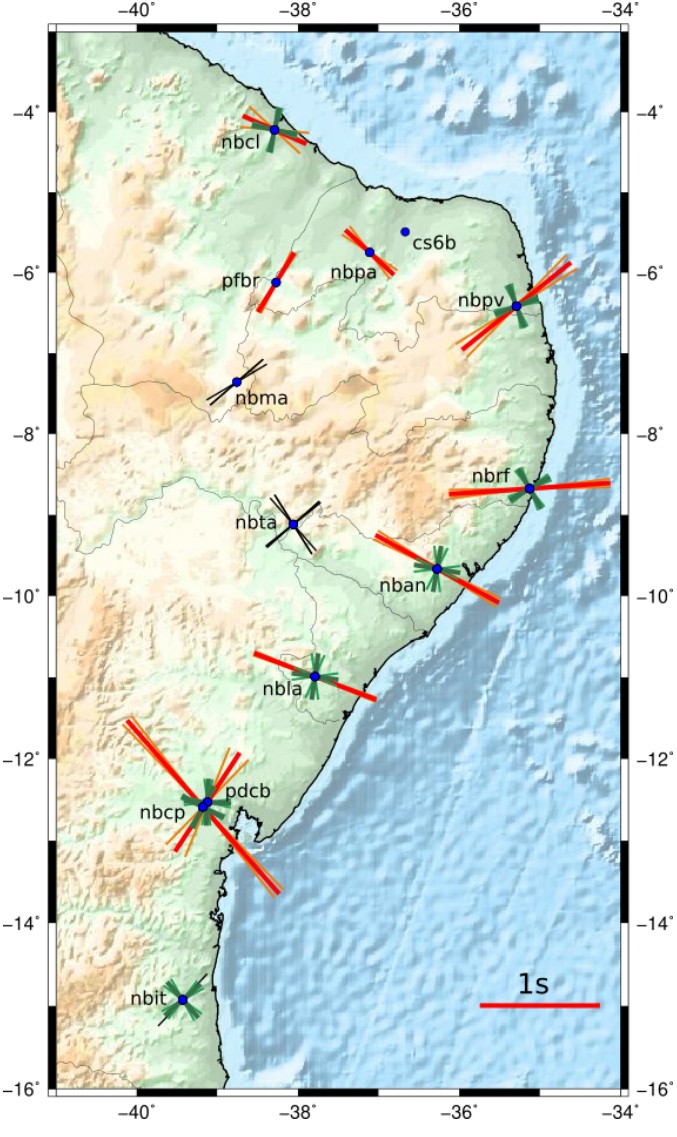

**Figure 7.** Comparison between fast axis of symmetry recorded by SKS-waves (red lines) and k=2 harmonics (green lines). SKS-splitting results are from Bastow et al. (2015); Assumpção et al. (2011). Red arrows refer to mean fast axis orientation (arrow direction) and delay time (arrow size) beneath the station. When SKS measurements provides only null measurements, we display black lines which are on the direction of the back-azimuth of the recorded event.

Deplaen et al. (2014), who argued that hotspot tectonism associated with the Mesozoic opening of the South Atlantic may have thermomechanically eroded Precambrian age fossil lithospheric fabrics beneath the CVL. Indeed, the presence of relatively shallow asthenosphere north-west of the Borborema Plateau has been recently postulated from a P-wave tomography study of the Borborema Province (Simões Neto et al., 2019). These authors identified a NE-SW trending low-velocity channel bordering





the Plateau that closely coincides with the observed isotropic alignment. Moreover, independent SKS splitting measurements performed at those stations by Bastow et al. (2015) reported either null measurements (stations nbma, cs6b, nbta) or really weak anisotropy (stations pfbr and nbpa).

## 6 Conclusions

We have investigated depth-dependent anisotropy in the Borborema Province of NE Brazil through harmonic decomposition of receiver functions developed at 39 stations in the region. Our main results include: (i) anisotropy within the Province is characterized by a horizontal to slightly dipping fast axis of symmetry; (ii) consistency of anisotropic orientations within the crust and the lithospheric mantle suggest a continuation of surface shear-zones down to lithospheric depths; (iii) fast axes of symmetry are oriented parallel to the main shear zones within the continental interior and sub-parallel to Mesozoic extension along the passive margins, consistent with a fossil origin inherited from the opening of the South Atlantic Ocean; (iv) absence of anisotropy along a NE-SW trending line in the center of the Province might be related to heating of the lithosphere by an asthenospheric channel identified in an independent tomography study.

*Acknowledgements.* Data used for this study were acquired due to funding from the national oil company Petrobras and the Conselho Nacional de Desenvolvimento Científico e Tecnológico (CNPq). GL was supported by a 1-year scholarship from the Programa Nacional de Pósdoutorado da Coordenação de Aperfeiçoamento de Pessoal de Nível Superior (PNPD/CAPES). This work was also supported by the "Laboratoire d'Excellence" LabexMer (ANR-10-LABX-19) and co-funded by a grant from the French government under the program "Investissements d'Avenir", and by a grant from the Regional Council of Brittany (SAD programme). JJ thanks the Conselho Nacional de Desenvolvimento Científico e Tecnológico (CNPq) for his research fellowship (CNPq, process no 304421/2015-4). We used the open-source toolbox GMT v.5.4 (Wessel et al., 2013) to produce the figures. Thanks are due to Nicola Piana Agostinetti for constructive discussions about harmonic decomposition method, and to Andrea Tommasi for interesting debates on the tectonics of northeast Brazil.



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
