# Peer review of "Lithospheric and sub-lithospheric deformation under the Borborema Province of NE Brazil from receiver function harmonic stripping"

_Solid Earth, 2019_

## Referee Comment (RC1) · Anonymous Referee #1 · 20 Mar 2019

Review of Lamarque and Julia, Solid Earth Discussions, 2019

This paper calculates teleseismic P-wave receiver functions to investigate the depth dependence of seismic anisotropy in the crust and lithospheric mantle in NE Brazil. The analysis considers the back-azimuth variations in observed receiver function signal and performs a harmonic decomposition to provide a quantitative estimate of anisotropy in terms of: 1) a plunging axis of symmetry and/or dipping interface; or 2) a horizontal axis of symmetry. The depth decomposition of the anisotropy is able to retrieve the average anisotropy in the crust and lithospheric mantle. The results show con-

sistent anisotropy in the crust and mantle, indicating a control by lithospheric-scale shear zones that develop during the Brasiliano-Pan African orogeny. The lack of well characterized anisotropy at some stations is taken as an indication of re-heating of the lithosphere by an asthenospheric channel. Stations along the Atlantic coast resolve fast anisotropic directions perpendicular to the margin, suggesting lithospheric inheritance during rifting.

General comments:

The paper compiles all available receiver function (RF) data and calculates new RF data for 11 recently installed stations. The RF analysis is adequately described and follows the standard procedures to obtain high-quality data. The novelty of this paper lies in the application of the harmonic decomposition to reveal depth-dependent anisotropy from back-azimuthal variations in the amplitude of both radial and tangential components of RF data, for individual stations. The results are discussed in an appropriate way, although part of the methodology lacks reference to original work that implemented variants of the technique (see specific comments). The condition for rejecting anisotropy (and therefore interpreting the subsurface structure as isotropic) could also be subject to debate. Overall the paper addresses an important question about the structure of fabrics beneath NE Brazil in relation with lithospheric inheritance and the significance of lithospheric-scale shear zones.

Specific comments:

The discussion of RF analysis is appropriate and includes proper referencing up to line 19 on page 7. There the authors describe an additional preliminary step in the harmonic decomposition analysis, where they migrate the time signals to depth using a 1D seismic velocity model to correct for the move-out of teleseismic waves. The migration to depth (before harmonic decomposition) was first proposed by Bianchi et al. (2010), who performed common-conversion (CCP) stacking using a dense line of stations and carried out the decomposition at CCP points. This method was further

applied in Piana Agostinetti et al. (2011) and Piana Agostinetti and Miller (2015). The step proposed here by the authors (converting time to depth at individual stations, as opposed to CCP stacking), was proposed by Audet (2015) and further applied in Cossette et al. (2016) and Tarayoun et al. (2017). The optimization of energy on one of the k=1 components (as done here) was also proposed by Audet (2015) to retrieve the dominant angle of anisotropy.

On page 10, the authors discuss the reliability of the anisotropic directions using a bootstrap analysis and consider that a measurement is unreliable if the bootstrap uncertainty is greater than 20 degrees. The bootstrap analysis returns an estimate of the standard error on the mean value on modeled parameters (such as the dominant angle of anisotropy), and confidence intervals are normally calculated from the standard error. Is this what is meant by "uncertainty" here? Is it 1-sigma (68% confidence) or 2-sigma (95%)? Furthermore, large variability in the recovered angle might not necessarily imply that the medium is isotropic. Strong structural heterogeneity might produce large-amplitude signal with apparent back-azimuth distribution with k>2. Alternatively, crystal symmetries might not always produce seismic anisotropy that can be modeled with the k=1 or k=2 components. So, it is still of interest to show the strength of the signal on the k=1 and k=2 energy components despite the large variability in bootstrap angles.

Following up from this comment, Figure 6 could be improved by plotting the relative amplitude of the corresponding energy components. On the maps, the anisotropy (length of bars) appears to be equal in magnitude at all stations, though I suspect that the energy components vary significantly from one station to another and regionally. This additional piece of information could also be included in the Discussion and compared with SKS splitting results.

Finally, it would be insightful to look at the receiver functions before application of the harmonic decomposition (e.g., in back-azimuth panels) to see why the "unreliable" stations have large uncertainty in anisotropic direction. This could be added to the Supplementary Information.

Technical corrections:

Page 8, line 3: "presents" -> present

Page 10, lines 6 and 10: the interval "[0,2]" ->. Do you mean [0, 2pi]?

Page 11, line 2: "Realize that" -> We note that

Page 12, line 3: "mantellic" -> mantle

Caption of Figure 6. It's not clear to me why the direction perpendicular to the fast axis is required in the case of horizontal anisotropy. Is it to differentiate between k=1 and k=2 directions? Which one of the two is the fast axis?

Page 14, line 14: "Bastow et al. (2011); Assumpçao et al. (2011)" -> Bastow et al. (2011) and Assumpçao et al. (2011)

Page 14, line 22: "sensible" -> sensitive

References (not appearing in paper):

Cossette et al.: Structure and anisotropy of the crust in the Cyclades, Greece, using receiver functions constrained by in situ rock textural data, J. Geophys. Res., 121, 2661-2678 (2016).

Piana Agostinetti et al.: Fluid migration in continental subduction: The Northern Apennines case study, Earth Planet. Sci. Lett., 302-267-278 (2011).

Piana Agostinetti and Miller: The fate of the downgoing oceanic plate: Insight from the Northern Cascadia subduction zone, Earth Planet. Sci. Lett., 408, 237-251 (2015).

Tarayoun et al.: Architecture of the crust and uppermost mantle in the northern Canadian Cordillera from receiver functions, J. Geophys. Res., 122, 5268-5287 (2017).

---

## Short Comment (SC1) · 2 Apr 2019

On lines 13 to 31 of p. 4, the authors acknowledge uplift in NE Brazil: "During the Mezosoic, opening of the south Atlantic separated the Borborema Province from its African conjugate (de Matos, 1992). Continental rifting resulted in significant crustal thinning in the region (Santos et al., 2014; Lima Neto et al., 2013; Luz et al., 2015b), forming both marginal (e.g. Ceará, Potiguar, Pernambuco-Paraíba, Sergipe-Alagoas) and intra-continental (e.g. Araripe, Tucano) sedimentary basins (Figure 1). The post Gondwana breakup evolution of the Borborema Province is characterized by recurrent

magmatism (Knesel et al., 2011) and postulated episodes of uplift in the Borborema Plateau (Morais Neto et al., 2009; de Oliveira and Medeiros, 2012) and the Araripe basin (Assine, 2007; Marques et al., 2014) ... Cenozoic uplift was inferred from both relative dating of elevated sediments of the Serra dos Martins formation in the northern Borborema Province and absolute dating from apatite fission-track analysis of granitic-gneissic samples (Morais Neto et al., 2009; de Oliveira and Medeiros, 2012). Although some sort of tectonic uplift and/or inversion of the Araripe basin seems to be widely accepted (Marques et al., 2014; Peulvast and Bétard, 2015; Garcia et al., 2019), uplift in the Borborema Plateau is more debated." Besides the evidence of inversion in the Araripe Basin, there is more evidence of inversion and uplift in NE Brazil. Coblentz and Richardson (1996) showed that the South American continent is under approximate E-W compression, which can lead to the inversion of previous structures favourably oriented. Current seismicity (Assumpção, 1998; Bezerra and Vita-Finzi, 2000; Bezerra et al., 2007, 2011, 2014; Ferreira et al., 1987, 1998, 2008; Lopes et al., 2010; Neto et al., 2013; Takeya et al., 1989) and deformation of Pleistocene or later sediments in the Taubaté Basin (Riccomini et al., 1989) provide evidence for ongoing compression in NE Brazil. Riccomini et al. (1989) and Cobbold et al. (2001) concluded that inversion(?) uplift(?) of the Brazilian Plateau (tectonic uplift of the eastern Brazilian margin, with greater expression at Serra do Mar close to the coast) is likely due to neotectonic activity. Morais Neto et al. (2009), Cogné et al. (2011, 2012, 2013) and Japsen et al. (2012) used AFTA to analyse episodic burial and exhumation in NE Brazil after the opening of the South Atlantic. Cogné et al. (2012), in particular, concluded that there has been synchronicity of the cooling phases in NE Brazil with the Steinman's Andean phases of tectonic uplift, and assumed a plate-wide compressional stress that reactivated inherited structures. Hegarty et al. (2002) used AFTA for a more local study, i.e. to analyse episodic burial and exhumation in the Araripe Basin. The reported results show two cooling events: (1) one at ca. 100–90 Ma, and (2) another from 30 to 0 Ma. The above evidence means that the latest stage of inversion is Quaternary, consistent with the works of Nóbrega et al. (2005), Morais Neto et al. (2009), Cogné

et al. (2011, 2012, 2013), Japsen et al. (2012), Gurgel et al. (2013) and Nogueira et al. (2015).

All this evidence of the stress state in South America and the inversion of inherited structures should be given in the Introduction of the ms. to adequately write the rationale of the paper.

References Assumpção, M., 1998. Seismicity and stresses in the Brazilian passive margin. SSA Bull. 88, 160–169. Bezerra, F.H.R., do Nascimento, A.F., Ferreira, J.M., Nogueira, F.C., Fuck, R.A., Brito Neves, B.B., Sousa,M.O.L., 2011. Review of active faults in the Borborema Province, intraplate South America integration of seismological and paleoseismological data. Tectonophysics 510, 269–290. Bezerra, F.H.R., Rossetti, D.F., Oliveira, R.G., Medeiros, W.E., Brito Neves, B.B., Balsamo, F., Nogueira, F.C.C., Dantas, E.L., Aandrades Filho, C., Góes, A.M., 2014. Neotectonic reactivation of shear zones and implications for faulting style and geometry in the continental margin of NE Brazil. Tectonophysics 614, 78–90. Bezerra, F.H.R., Takeya,M.K., Sousa, M.O.L., Do-Nascimento, A.F., 2007. Coseismic reactivation of the Samambaia fault. Tectonophysics 430, 27–39. Bezerra, F.H.R., Vita-Finzi, C., 2000. How active is a passive margin? Paleoseismicity in northeastern Brazil. Geology 28, 591–594. Cobbold, P.R., Meisling, K.E., Mount, V.S., 2001. Reactivation of an obliquely rifted margin, Campos and Santos Basins, southeastern Brazil. AAPG Bull. 85, 1925–1944. Coblentz, D.D., Richardson, R.M., 1996. Analysis of the South American intraplate stress field. J. Geophys. Res. 101, 8643–8657. Cogné, N., Cobbold, P.R., Riccomini, C., Gallagher, K., 2013. Tectonic setting of the Taubaté Basin (southeastern Brazil): insights from regional seismic profiles and outcrop data. J. S. Am. Earth Sci. 42, 194–204. Cogné, N., Gallagher, K., Cobbold, P.R., 2011. Post-rift reactivation of the onshore margin of southeast Brazil: evidence from apatite (U–Th)/He and fission-track data. Earth Planet. Sci. Lett. 309, 118–130. Cogné, N., Gallagher, K., Cobbold, P.R., Riccomini, C., Gautheron, C., 2012. Post-breakup tectonics in southeast Brazil from thermochronological data and combined inverse forward thermal history modelling. J.

Geophys. Res. 117, B11413. http://dx.doi.org/10.1029/2012JB009340. Ferreira, J.M., Bezerra, F.H.R., Sousa,M.O.L., Do Nascimento, A.F., Sa, J.M., Franca, G.S., 2008. The role of Precambrian mylonitic belts and present-day stress field in the coseismic reactivation of the Pernambuco lineament, Brazil. Tectonophysics 456, 111–126. Ferreira, J.M., Oliveira, R.T., Takeya, M.K., Assumpção, M., 1998. Superposition of local and regional stresses in northeast Brazil: evidence from focal mechanisms around the Potiguar marginal basin. Geophys. J. Int. 134, 341–355. Ferreira, J.M., Takeia,M., Costa, J.M.,Moreira, J.A., Assumpção,M., Veloso, J.A.V., Pearce, R.G., 1987. A continuing intraplate earthquake sequence near João Câmara, Northeastern Brazil — preliminary results. Geophys. Res. Lett. 14, 1402–1405. Gurgel, S.P.P., Bezerra, F.H.R., Corrêa, A.C.B., Marques, F.O., Maia, R.P., 2013. Cenozoic uplift and erosion of structural landforms in NE Brazil. Geomorphology 186, 68-84. Hegarty, K.A., Morais Neto, J.M., Karner, G.D., 2002. The enigma of the Araripe Plateau: new constraints on its uplift and tectonic history using AFTA. XLI Congresso Brasileiro de Geologia, Extended Abstract. Japsen, P., Bonow, J., Green, P.F., Cobbold, P.R., Chiossi, D., Lilletveit, R., Magnavita, L.P., Pedreira, A., 2012. Episodic burial and exhumation in NE Brazil after opening of the South Atlantic. GSA Bull. 124, 800–816. Lopes, A.E.V., Assumpção, M., do Nascimento, A.F., Ferreira, J.M., Menezes, E.A.S., Barbosa, J.R., 2010. Intraplate earthquake swarm in Belo Jardim, NE Brazil: reactivation of a major Neoproterozoic shear zone (Pernambuco Lineament). Geophys. J. Int. 180, 1303–1312. Morais Neto, J.M., Hegarty, K.A., Karner, G.D., Alkmim, F.F., 2009. Timing and mechanisms for the generation and modification of the anomalous topography of the Borborema Province, northeastern Brazil. Mar. Pet. Geol. 26, 1070–1086. Neto, H.C.L., Ferreira, J.M., Bezerra, F.H.R., Assumpção, M.S., do Nascimento, A.F., Sousa, M.O.L., Menezes, E.A.S., 2013. Upper crustal earthquake swarms in São Caetano: reactivation of the Pernambuco shear zone and trending branches in intraplate Brazil. Tectonophysics 608, 804–811. Nóbrega, M.A., Sá, J.M., Bezerra, F.H.R., Hadler Neto, J.C., Iunes, P.J., Guedes, S., Tello Saenz, C.A., Hackspacher, P.C., Lima-Filho, F.P., 2005. The use of apatite fission track thermochronology to constrain fault movement

and sedimentary basin evolution in northeastern Brazil. Radiat. Meas. 39, 627–633. Nogueira, F.C.C., Marques, F.O., Bezerra, F.H.R., de Castro, D.L., Fuck, R.A., 2015. Cretaceous intracontinental rifting and post-rift inversion in NE Brazil: insights from the Rio do Peixe Basin. Tectonophysics 644-645, 92-107. Reis, A.F.C., Bezerra, F.H.R., Ferreira, J.M., do Nascimento, A.F., Lima, C.C., 2013. Stress magnitude and orientation in the Potiguar Basin, Brazil: implications on faulting style and reactivation. J. Geophys. Res. Solid Earth 118, 1–14. http://dx.doi.org/10.1002/2012JB009953. Riccomini, C., Peloggia, A.U.G., Saloni, J.C.L., Kohnke, M.W., Figueira, R.M., 1989. Neotectonic activity in the Serra do Mar rift system (southeastern Brazil). J. S. Am. Earth Sci. 2, 191–197. Takeya, M.K., Ferreira, J.M., Pearce, R.P., Assumpção,M., Costa, J.M., Sophia, C.M., 1989. The 1986–1987 intraplate earthquakes sequence near João Câmara, northeast Brazil — evolution of seismicity. Tectonophysics 167, 117–131.

F.O. Marques, F.H. Bezerra, F.C. Nogueira

---

## Referee Comment (RC2) · Anonymous Referee #2 · 5 Apr 2019

The manuscript under review presents a detailed accounting of lithospheric anisotropy through the use of Ps receiver function analysis and data collected at 75 seismic stations within the Borborema Province of NE Brazil. The importance of their analysis rests in the fact that they can provide firm constraints on anisotropic boundary depth, in contrast to shear wave splitting which is a path integrated measurement. Their results show a clear correlation between tectonic deformation and orientation of seismic anisotropy. Within the continent, they find that the orientation of anisotropy is coincident with the orientation of large-scale shear zones thought to be associated with

the Brasiliano-Pan African Orogeny. On the coast, anisotropy is oriented perpendicular to the coastline, suggesting that rifting is the process responsible for generating anisotropy. In places where anisotropy is absence, it is inferred that heating by the asthenosphere may have destroyed any preexisting lithospheric fabric.

Comments regarding methodology: Overall, the methodology is thoroughly and carefully described, and proper citations were given. My only question is regarding the cut-off for the minimum number of bins with data (lines 24-26). The authors require a minimum of 9 bins with data (90 degrees), which can be either continuous or discontinuous. Why was this minimum chosen? Is there an appreciable difference in how well the harmonic decomposition works? Do the authors have synthetic example they could show to demonstrate their reasoning? The reason I ask is because this seems to be the primary reason for reducing the number of stations from 75 to 39.

Comments regarding results: I appreciated the inclusion of the harmonically decomposed results within the supplementary materials. They clearly exhibit evidence of anisotropy. I did however wonder how the authors dealt with cases where more than one anisotropic boundary was present within either the crust or the mantle. I may have missed where they spoke to this, but could not find it upon reexamining the manuscript. A clearer description would have been greatly appreciated.

Comments regarding interpretation: My only significant concern with the manuscript was that while regional patterns of deformation matched the fast direction, it was not always clear to me that the material properties would necessitate such an answer. For example, while the LPO of olivine typically means that the A-axis of olivine is oriented in the same direction as strain, the crust is significantly more complex, as several candidate minerals can generate different types of anisotropy, in addition to the possibility of shape preferred orientation of different materials. I would encourage the authors to think more carefully about crustal anisotropy in particular.

Comments regarding figures: Figure 6: It would be useful if the names of the stations

were more clearly written as they appear washed out and are difficult to read.

---

## Author Comment (AC3) · 30 Apr 2019

We greatly thank Fernando Ornelas Marques for such detailed references about inversion and uplift in South American continent and in NE Brazil. We do our best to integrate these informations within our geological settings by citing local studies on NE Brazil, which is our region of interest in this paper.

Modifications within the manuscriptÂă:

[Figure]

lines 27 to 31 page 4Ăă:

"Cenozoic uplift along the northeastern brazilian margin was inferred from relative dating of elevated sediments of the Serra dos Martins formation absolute dating from apatite fission-track analysis of granitic-gneissic and sedimentary samples and geomorphological studies (Morais Neto et al., 2009; de Oliveira and Medeiros, 2012, Nóbrega et al., 2005Ăă; Nogueira et al., 2015). Although some sort of tectonic uplift and/or inversion of the Araripe basin seems to be widely accepted (Marques et al., 2014; Peulvast and Bétard, 2015; Garcia et al., 2019), uplift in the Borborema Plateau is more debated"

---

## Editor Comment (EC1) · Caroline Beghein (Editor) · 7 May 2019

Dear author,

Thank you for the thorough revisions to your manuscript. I would love to see the interpretation justification that you wrote in the response to reviewer 2 included in the manuscript itself: "A complex combination of LPO and SPO could be present in the mantle, although LPO is likely to dominate (Nicolas and Christensen, 1987; Silver 1996; Mainprice et al., 2000); fractures and cracks or fine layering, could addition-

ally contribute in the crust. For that reason, our interpretations focus dominantly on mantle anisotropy, consistency of anisotropy within the lithosphere (crust and mantle), and regional-scale trends. And, to avoid a bias related to local features, we refrain from interpreting small-scale variations in anisotropy within the crust."

Please, include those few sentences in your paper. Thank you.

Caroline Beghein

---

## Author Comment (AC4) · 8 May 2019

Dear Editor,

We will add the interpretation justification at the beginning of our discussion.
* * *

---

## Editor Comment (EC2) · Caroline Beghein (Editor) · 12 May 2019

Dear Dr Lamarque,

The latest version of the manuscript that I have access to (version 4) does not seem to include those last changes. Could you please make sure they are included and upload a new version before Monday 13 May? Thank you.

Caroline Beghein

---

## Author Response (AR1)

Answers to Anonymous Referee 1

In order to make answers as clear as possible, we copy in black the reviewer comments, and answer with blue color. Proposed changes in the original text are indicated in red.

Anonymous Referee #1

Review of Lamarque and Julia, Solid Earth Discussions, 2019

This paper calculates teleseismic P-wave receiver functions to investigate the depth dependence of seismic anisotropy in the crust and lithospheric mantle in NE Brazil. The analysis considers the back-azimuth variations in observed receiver function signal and performs a harmonic decomposition to provide a quantitative estimate of anisotropy in terms of: 1) a plunging axis of symmetry and/or dipping interface; or 2) a horizontal axis of symmetry. The depth decomposition of the anisotropy is able to retrieve the average anisotropy in the crust and lithospheric mantle. The results show consistent anisotropy in the crust and mantle, indicating a control by lithospheric-scale shear zones that develop during the Brasiliano-Pan African orogeny. The lack of well characterized anisotropy at some stations is taken as an indication of re-heating of the lithosphere by an asthenospheric channel. Stations along the Atlantic coast resolve fast anisotropic directions perpendicular to the margin, suggesting lithospheric inheritance during rifting.

General comments:

The paper compiles all available receiver function (RF) data and calculates new RF data for 11 recently installed stations. The RF analysis is adequately described and follows the standard procedures to obtain high-quality data. The novelty of this paper lies in the application of the harmonic decomposition to reveal depth-dependent anisotropy from back-azimuthal variations in the amplitude of both radial and tangential components of RF data, for individual stations. The results are discussed in an appropriate way, although part of the methodology lacks reference to original work that implemented variants of the technique (see specific comments). The condition for rejecting anisotropy (and therefore interpreting the subsurface structure as isotropic) could also be subject to debate. Overall the paper addresses an important question about the structure of fabrics beneath NE Brazil in relation with lithospheric inheritance and the significance of lithospheric-scale shear zones.

We greatly thank the reviewer for a detailed reading of our manuscript. In particular, we appreciate the remark on our interpretation of the stations with large variability in anisotropic parameters, which helped us improve our interpretation of the results and their geodynamic implications.

Specific comments:

The discussion of RF analysis is appropriate and includes proper referencing up to line 19 on page 7. There the authors describe an additional preliminary step in the harmonic decomposition analysis, where they migrate the time signals to

depth using a 1D seismic velocity model to correct for the move-out of teleseismic waves. The migration to depth (before harmonic decomposition) was first proposed by Bianchi etal. (2010), who performed common-conversion (CCP) stacking using a dense line of stations and carried out the decomposition at CCP points. This method was further applied in Piana Agostinetti et al. (2011) and Piana Agostinetti and Miller (2015). The step proposed here by the authors (converting time to depth at individual stations, as opposed to CCP stacking), was proposed by Audet (2015) and further applied in Cossette et al. (2016) and Tarayoun et al. (2017). The optimization of energy on one of the k=1 components (as done here) was also proposed by Audet (2015) to retrieve the dominant angle of anisotropy.

The reviewer is correct when pointing out that migration before harmonic stripping was already proposed in previous works. We have added the missing references to the updated manuscript. The proposed change in the text is:

« Prior to implementing the anisotropy analysis, each radial and tangential receiver function was migrated to depth after P to S ray-tracing through the global velocity model ak135-f (Kennett et al., 1995; Montagner and Kennett, 1996). The purpose of the migration is to correct the phase move-out introduced by varying incidence angles among the incoming teleseismic P-wavefronts, effectively equalizing the receiver function waveforms in the depth domain (Dueker and Sheehan, 1997). Migration before harmonic stripping at individual stations was previously utilized by Audet (2015), Causette et al. (2016) and Tarayoun et al. (2017). Similarly, Bianchi et al (2010), Piana Agostinetti et al (2011), and Piana Agostinetti and Miller (2015) applied harmonic decomposition on depth-migrated cross-sections obtained through CCP stacking of receiver functions. Next, the migrated radial and transverse receiver functions for each station were grouped by back-azimuth in 36 non-overlapping, ...»

On page 10, the authors discuss the reliability of the anisotropic directions using a bootstrap analysis and consider that a measurement is unreliable if the bootstrap uncertainty is greater than 20 degrees. The bootstrap analysis returns an estimate of the standard error on the mean value on modeled parameters (such as the dominant angle of anisotropy), and confidence intervals are normally calculated from the standard error. Is this what is meant by "uncertainty" here? Is it 1-sigma (68% confidence) or 2-sigma (95%)?

Uncertainties refer to the 2-sigma standard error obtained from a population of 200 angle estimates developed from bootstrapping the original dataset. Additional text will be added to the manuscript to clarify that point:

« In order to estimate uncertainties, we applied a bootstrap statistical approach by randomly re-sampling with replacement our receiver functions. We performed such analysis with 200 replications at each of the selected stations. From these 200 values, we estimated the standard error (2-sigma), which corresponds to the uncertainty in the direction of the fast-axis of symmetry. A measurement is considered as not reliable, and then rejected, if the estimated uncertainties are larger than 20°.

Furthermore, large variability in the recovered angle might not necessarily imply that the medium is isotropic. Strong structural heterogeneity might

produce large-amplitude signal with apparent back-azimuth distribution with k>2. Alternatively, crystal symmetries might not always produce seismic anisotropy that can be modeled with the k=1 or k=2 components. So, it is still of interest to show the strength of the signal on the k=1 and k=2 energy components despite the large variability in bootstrap angles. Following up from this comment, Figure 6 could be improved by plotting the relative amplitude of the corresponding energy components. On the maps, the anisotropy (length of bars) appears to be equal in magnitude at all stations, though I suspect that the energy components vary significantly from one station to another and regionally. This additional piece of information could also be included in the Discussion and compared with SKS splitting results. Finally, it would be insightful to look at the receiver functions before application of the harmonic decomposition (e.g., in back-azimuth panels) to see why the "unreliable" stations have large uncertainty in anisotropic direction. This could be added to the Supplementary Information.

The reviewer is correct that large variability in the recovered angles at «unreliable» stations is not necessarily related to weak anisotropy under those stations. Following her/his advice, we have now calculated the energy and inspected transverse component amplitudes in detail. We found that energy at the "unreliable" stations is as strong as that found at the "reliable" ones (see results for station cs6b in the additional supplementary material). To make this clear, Figure 6 has been updated to display energy level at each station (for clarity reasons, we prefer to keep constant bar lengths and denote energy through color-coding the station symbol). The new legend for Figure 6 will be:

« A) Map of symmetry directions (dark lines) obtained for the crust (0-32 km). When one line is plotted at the station, it represents either the trend of the dip, in the case of dipping interface, or the trend of the fast axis in the case of plunging anisotropy. When two lines are plotted, they refer to the fast axis and to its perpendicular direction for horizontal anisotropy. Light colors represent $2\sigma$ uncertainties estimated from the bootstrap (after re-sampling 200 times). B) Same as for the lithospheric mantle (32-100 km). Station symbols have been color-coded according to the energy level of the dominant harmonic degree. »

It is now clear that our original interpretation of large angle variability as resulting from a weak anisotropic signature under the station was incorrect. We agree with the reviewer that non-azimuthal anisotropy and/or strong structural heterogeneities provide a more likely explanation. In any case, this non-azimuthal anisotropy is still located above a NE-SW trending channel of thin lithosphere and shallow asthenosphere. Accordingly, we now propose that deformation from thermo-mechanical erosion by horizontal, sub-lithospheric flow - previously postulated in the tomographic study of Simões Neto et al. (2019) - must be ongoing above the NE-SW channel. Also, as initial thinning of the lithosphere along the channel was probably triggered by Mesozoic extension along the Cariri-Potiguar trend, alterations of the original Precambrian anisotropic fabric by Mesozoic extension might still be present above the channel. Additionally, we note that the location of the Cariri-Potiguar trend also marks the boundary between the EW striking shear zones in the southern Province from the NE-SW striking shear zones in the western Province (Figure 1). This suggests the Cariri-Potiguar trend also marks the location of a

former paleo-suture that later acted as a zone of weakness along which the Mesozoic rift (now aborted) developed.

Thus, we believe the non-azimuthal anisotropy recorded at stations located along this trend is more likely related to complex fossil anisotropic fabrics resulting from a combination of deformation along the ancient collision between Precambrian blocks, Mesozoic extension, and thermo-mechanical erosion/mantle dragging by sub-lithospheric flow.

Modifications within the manuscript :
«
5.3. Non-azimuthal anisotropy along the aborted Cariri-Potiguar rift
At a number of stations (ar05, nbma, pfbr, nbpa, cs6b), uncertainties for the direction of the fast axis of anisotropy are larger than 20°. ~~We think that anisotropy is just too small to be confidently retrieved, and interpret those stations as sampling an isotropic lithosphere. These stations record signal on the tranverse component (see example for station cs6b in supplementary materials, Figure S3) , indicating the presence of anisotropy at depth. Energy on k=1 or k=2 is of similar intensity to the energy at stations with smaller uncertainties, as displayed in Figure 6.One explanation for the absence of lithospheric-scale anisotropy could be the destruction of anisotropic fabrics through sub-lithospheric heating of the overlying lithosphere. This hypothesis was proposed for the Cameroon Volcanic Line (CVL) by Deplaen et al. (2014), who argued that hotspot tectonism associated with the Mesozoic opening of the South Atlantic may have thermomechanically eroded Precambrian age fossil lithospheric fabrics beneath the CVL. Indeed, the presence of relatively shallow asthenosphere north-west of the Borborema Plateau has been recently postulated from a P-wave tomography study of the Borborema Province (Simões Neto et al., 2019). These authors identified a NE-SW trending low-velocity channel bordering the Plateau that closely coincides with the observed isotropic alignment. Moreover, independent SKS splitting measurements performed at those stations by Bastow et al. (2015) reported either null measurements (stations nbma, cs6b, nbta) or really weak anisotropy (stations pfbr and nbpa).~~ This NE-SW oriented line is located above a NE-SW trending channel of thin lithosphere imaged by the tomographic study of Simões Neto et al. (2019). We suggest that deformation from thermo-mechanical erosion by horizontal, sub-lithospheric flow along the channel - also postulated by Simões Neto et al. (2019) - must be ongoing above this NE-SW channel. Also, as initial thinning of the lithosphere along the channel was triggered by Mesozoic extension along the Cariri-Potiguar trend, alterations to the original Precambrian anisotropic fabric by Mesozoic extension might still be present. Additionally, we note that the location of the Cariri-Potiguar trend also marks the boundary between the EW striking shear zones in the southern Province from the NE-SW striking shear zones in the western Province (Figure 1). This suggests the Cariri-Potiguar trend also marks the location of a former paleo-suture that later acted as a zone of weakness along which the Mesozoic rift (now aborted) could develop. Thus, we believe the non-azimuthal anisotropy recorded at stations located along this trend is likely related to complex fossil anisotropic fabrics resulting from a combination of deformation along the ancient collision between Precambrian blocks, Mesozoic extension, and thermo-mechanical erosion/mantle dragging by sub-lithospheric flow.  »

Technical corrections:

Page 8, line 3: "presents" -> present.
Done.

Page 10, lines 6 and 10: the interval "[0,2]" ->. Do you mean [0, 2pi]?
Yes, it's been modified accordingly.

Page 11, line 2: "Realize that" -> We note that
Done.

Page 12, line 3: "mantellic" -> mantle
Done.

Caption of Figure 6. It's not clear to me why the direction perpendicular to the fast axis is required in the case of horizontal anisotropy. Is it to differentiate between k=1 and k=2 directions? Which one of the two is the fast axis?

In the case of anisotropy with pure horizontal fast axis of symmetry, the energy is only on the k=2 harmonics and receiver functions display a 4-lobed back-azimuthal pattern. A synthetic example of that case is visible in Schulte Pelkum and Mahan (2014), Figures 2a and 3a. This 4-lobed back-azimuthal pattern implies maximum amplitudes for 4 directions, which correspond to: (i) the direction of the fast axis of symmetry, (ii) the direction opposite to the fast-axis of symmetry, (iii) the direction perpendicular to the fast axis of symmetry, and, (iv) the direction opposite to the perpendicular. It is thus not possible to discriminate between the fast axis of symmetry and the direction perpendicular to it through analysis of the k=2 harmonics.

Page 14, line 14: "Bastow et al. (2011); Assumpçao et al. (2011)" -> Bastow et al.(2011) and Assumpçao et al. (2011)
Done.

Page 14, line 22: "sensible" -> sensitive
Done.

References (not appearing in paper):
Cossette et al.: Structure and anisotropy of the crust in the Cyclades, Greece, using receiver functions constrained by in situ rock textural data, J. Geophys. Res., 121,2661-2678 (2016).
Piana Agostinetti et al.: Fluid migration in continental subduction: The Northern Apen-nines case study, Earth Planet. Sci. Lett., 302-267-278 (2011).
Piana Agostinetti and Miller: The fate of the downgoing oceanic plate: Insight from the Northern Cascadia subduction zone, Earth Planet. Sci. Lett., 408, 237-251 (2015).
Tarayoun et al.: Architecture of the crust and uppermost mantle in the northern Canadian Cordillera from receiver functions, J. Geophys. Res., 122, 5268-5287 (2017).

All the missing references have now been added to the reference list.

In order to make answers as clear as possible, we copy in black the reviewer comments, and answer with blue color. Proposed changes in the original text are indicated in red.

Anonymous Referee #2

The manuscript under review presents a detailed accounting of lithospheric anisotropy through the use of Ps receiver function analysis and data collected at 75 seismic stations within the Borborema Province of NE Brazil. The importance of their analysis rests in the fact that they can provide firm constraints on anisotropic boundary depth, in contrast to shear wave splitting which is a path integrated measurement. Their results show a clear correlation between tectonic deformation and orientation of seismic anisotropy. Within the continent, they find that the orientation of anisotropy is coincident with the orientation of large-scale shear zones thought to be associated with the Brasiliano-Pan African Orogeny. On the coast, anisotropy is oriented perpendicular to the coastline, suggesting that rifting is the process responsible for generating anisotropy. In places where anisotropy is absence, it is inferred that heating by the asthenosphere may have destroyed any preexisting lithospheric fabric.

Comments regarding methodology: Overall, the methodology is thoroughly and carefully described, and proper citations were given. My only question is regarding the cut-off for the minimum number of bins with data (lines 24-26). The authors require a minimum of 9 bins with data (90 degrees), which can be either continuous or discontinuous. Why was this minimum chosen? Is there an appreciable difference in how well the harmonic decomposition works? Do the authors have synthetic example they could show to demonstrate their reasoning? The reason I ask is because this seems to be the primary reason for reducing the number of stations from 75 to 39.

The reason for selecting stations that display data in at least 9 bins (10 degrees wide) is purely geometrical. Recall that we use receiver functions to map anisotropy with either 2-lobed (plunging fast axis of symmetry) or 4-lobed (horizontal fast axis of symmetry) back-azimuthal patterns. In the case of a plunging fast axis of symmetry, 9 bins corresponds to half the period for a 2-lobed pattern (90 degrees); in the case of a horizontal fast axis of symmetry, 9 bins corresponds to a full period for a 4-lobed pattern (90 degrees). By requiring 9 bin coverage (10 degree wide), we are able to reliably display either a 2-lobed or a 4-lobed pattern.

We propose to modify the text in the manuscript as:

« Next, the migrated radial and transverse receiver functions for each station were grouped by back-azimuth in 36 non-overlapping, 10° wide bins, and averaged within each bin. A given station was then selected if it presented at least two averaged receiver functions (one radial and one tangential) in at least 9 bins. This selection criterion ensured a sampling of at least 90° in back-azimuth, either continuously or discontinuously, around the station. A back-azimuthal coverage

from at least 9 bins (each 10° wide) allows the mapping of either half the period for a 2-lobed pattern (anisotropy with plunging fast axis of symmetry) or a full period for a 4-lobed pattern (anisotropy with horizontal fast axis of symmetry). A total of 39 stations were thus selected for anisotropy analysis. An example of stacked and migrated receiver functions is displayed in Figure 4. »

Comments regarding results: I appreciated the inclusion of the harmonically decomposed results within the supplementary materials. They clearly exhibit evidence of anisotropy. I did however wonder how the authors dealt with cases where more than one anisotropic boundary was present within either the crust or the mantle. I may have missed where they spoke to this, but could not find it upon reexamining the manuscript. A clearer description would have been greatly appreciated.

Our goal in this paper is to examine the direction of the dominant anisotropy within two depth windows, which correspond to the crust and the lithospheric mantle. We make the assumption that in the case of several anisotropic layers, the layer with the strongest anisotropy will dominate the results. We are aware that results can reflect the average value from different anisotropic layers, or from different types of anisotropy in the case of similar anisotropic strength.

We propose to modify the text in the manuscript as:

"4. Results
Anisotropy parameters were examined for each station at two depth-window ranges: (1) crust (Figure 6A), which was assumed to be located between 0 and 33 km depth, in agreement with the 32-40 km range estimated by Luz et al. (2015b) under the Borborema Plateau and 30-33 km under the surrounding basins; and (2) lithospheric mantle, which was taken to be between 33 and 100 km depth (Figure 6B). We assume that the layer with the strongest anisotropy will dominate the results in the case of several anisotropic layers. However, it might happen that results reflect the average value from different anisotropic layers, or from different types of anisotropy in the case of similar anisotropic strength. All results are indicated in Table 1.
An inspection of Figure 6A reveals that the crust of northeast Brazil …"

Comments regarding interpretation: My only significant concern with the manuscript was that while regional patterns of deformation matched the fast direction, it was not always clear to me that the material properties would necessitate such an answer. For example, while the LPO of olivine typically means that the A-axis of olivine is oriented in the same direction as strain, the crust is significantly more complex, as several candidate minerals can generate different types of anisotropy, in addition to the possibility of shape preferred orientation of different materials. I would encourage the authors to think more carefully about crustal anisotropy in particular.

We agree with this remark. A complex combination of LPO and SPO could be present in the mantle, although LPO is likely to dominate (Nicolas and Christensen, 1987; Silver 1996; Mainprice et al., 2000); fractures and cracks or fine layering, could additionally contribute in the crust. For that reason, our

interpretations focus dominantly on mantle anisotropy, consistency of anisotropy within the lithosphere (crust and mantle), and regional-scale trends. And, to avoid a bias related to local features, we refrain from interpreting small-scale variations in anisotropy within the crust.

Comments regarding figures: Figure 6: It would be useful if the names of the stations were more clearly written as they appear washed out and are difficult to read.
Done.

References :
* Mainprice D., Barruol G.. Ben Ismaïl W.. Karato S.-I., Forte A.,  Liebermann R., Masters G.,  Stixrude L.. 
[revised manuscript text omitted]

Figure S3 is an example of recorded radial and transverse receiver functions at station CS6B. Receiver functions are plotted as a function of back-azimuth. The transverse component record amplitude for mantle depths but no periodic pattern is visible.

15  This "non-azimuthal" anisotropy has similar energy than azimuthal anisotropy recorded at other stations (see Figure 6).

[Figure]

[Figure]

[Figure]

[Figure]

[Figure]

[Figure]

[Figure]

[Figure]

[Figure]

[Figure]

harmonics

[Figure]

energy

[Figure]

[Figure]

**Figure S2.** Energy on k=1 and k=2 harmonics are calculated for synthetics within a velocity model with 3 layers and constant Vp and Vs. The first and third layers are isotropic whereas the second layer display 6% anisotropy for both P and S-waves with fast axis dipping indicated above each corresponding graph (0, 10, 15 and 20°).

[Figure]

**Figure S3.** Radial and transverse receiver functions at station CS6B